

**Formation drivers and photochemical effects of ClNO₂ in a coastal city of**
**Southeast China**
Gaojie Chen[1,4,5], Xiaolong Fan[1,4], Haichao Wang[2*], Yee Jun Tham[3], Ziyi Lin[1,4,5], Xiaoting Ji[1,4,5], Lingling
Xu[1,4], Baoye Hu[6], Jinsheng Chen[1,4*]
[1]Center for Excellence in Regional Atmospheric Environment, Institute of Urban Environment, Chinese
Academy of Sciences, Xiamen 361021, China
[2]School of Atmospheric Sciences, Sun Yat-sen University, Zhuhai 519082, China
[3]School of Marine Sciences, Sun Yat-sen University, Zhuhai 519082, China
[4]Fujian Key Laboratory of Atmospheric Ozone Pollution Prevention, Institute of Urban Environment,
Chinese Academy of Sciences, Xiamen 361021, China
[5]University of Chinese Academy of Sciences, Beijing 100049, China
[6]Minnan Normal University, Zhangzhou 363000, China
*Correspondence to: Jinsheng Chen (jschen@iue.ac.cn); Haichao Wang (wanghch27@mail.sysu.edu.cn).





**Abstract.** Nitryl chloride (ClNO$_2$) is an important precursor of chlorine (Cl) radical, significantly affecting ozone (O$_3$) formation and photochemical oxidation. However, the key drivers of ClNO$_2$ production are not fully understood. In this study, the field observations of ClNO$_2$ and related parameters were conducted in a coastal city of Southeast China during the autumn of 2022, combining with machine learning and model simulations to elucidate its key influencing factors and atmospheric impacts. Elevated concentrations of ClNO$_2$ (> 500 ppt) were notably observed during nighttime in late autumn, accompanied by increased levels of dinitrogen pentoxide (N$_2$O$_5$) and nitrate (NO$_3^-$). Nighttime concentrations of ClNO$_2$ peaked at 3.4 ppb, while its daytime levels remained significant, reaching up to 100 ppt and sustaining at approximately 40 ppt at noon. Machine learning and field observations identified nighttime N$_2$O$_5$ heterogeneous uptake as the predominant pathway for ClNO$_2$ production, whereas NO$_3^-$ photolysis contributed to its daytime generation. Additionally, ambient temperature (T) and relative humidity (RH) emerged as primary meteorological factors affecting ClNO$_2$ formation, mainly through their effects on thermal equilibrium and N$_2$O$_5$ hydrolysis processes, respectively. Ultraviolet (UV) radiation was found to play a dual role in ClNO$_2$ concentrations around noon. Box model simulations showed that under high ClNO$_2$ conditions, the rates of alkane oxidation by Cl radical in the early morning exceeded those by OH radical. Consequently, VOC oxidation by Cl radical contributed ~ 19% to RO$_x$ production rates, thereby significantly impacting O$_3$ formation and atmospheric oxidation capacity. This research enriched the understanding of ClNO$_2$ generation and loss pathways, providing valuable insights for the regulation of photochemical pollution in coastal regions.



59

**1 Introduction**

Chlorine (Cl) radical, as an important atmospheric oxidant, can react with volatile organic compounds (VOCs) to affect $RO_x$ (including OH, $HO_2$, and $RO_2$) radicals and ozone ($O_3$) formation (Yi et al., 2023), thereby perturbing atmospheric chemical components and air quality (Peng et al., 2021; Li et al., 2020). The reaction rates between Cl radical and some alkanes were several orders of magnitude faster than those involving OH radical (Atkinson et al., 2006). Furthermore, the related studies indicated that the production rates of Cl radical in the early morning could significantly exceed the production rates of OH radical formed via $O_3$ photolysis (Phillips et al., 2012; Tham et al., 2016), thereby enhancing the atmospheric oxidation capacity.

Nitryl chloride ($ClNO_2$) is one of the major Cl radical precursors in the tropospheric atmosphere (Thornton et al., 2010; Xue et al., 2015; Liu et al., 2017). It is mainly generated by the heterogeneous uptake of dinitrogen pentoxide ($N_2O_5$) on chloride-containing aerosols (Finlayson-Pitts et al., 1989; Thornton et al., 2010), among which $N_2O_5$ is produced through the equilibrium reaction with nitrogen dioxide ($NO_2$) and nitrate ($NO_3$) radical. Since Osthoff et al. (2008) firstly detected over 1 ppb of $ClNO_2$ in the urban outflows of America (Osthoff et al., 2008), significant production of $ClNO_2$ has been widely observed in the polluted coastal and inland areas with abundant anthropogenic emissions and chloride sources, with concentrations ranging from tenths of ppt to several ppb (Riedel et al., 2012; Mielke et al., 2013; Mielke et al., 2011; Phillips et al., 2012; Bannan et al., 2015; Wang et al., 2016; Xia et al., 2020; Xia et al., 2021; Yun et al., 2018a; Wang et al., 2022; Li et al., 2023). For the diurnal profile of $ClNO_2$, its concentrations generally peaked and accumulated at midnight, then rapidly decreased to low levels due to strong photolysis after sunrise (Ma et al., 2023; Mielke et al., 2011; Xia et al., 2020). However, elevated daytime concentrations of $ClNO_2$ have been observed in field studies, mainly attributed to reduced photolysis rates under heavy cloud or fog cover, as well as contributions from horizontal and vertical transport (Tham et al., 2016; Xia et al., 2021; Jeong et al., 2019; Mielke et al., 2013; Bannan et al., 2015). Notably, the recent laboratory research demonstrated that nitrate ($NO_3^-$) photolysis can generate $ClNO_2$ alongside $Cl_2$ (Dalton et al., 2023), yet this mechanism has not been confirmed under real atmospheric conditions.

At present, the observation studies of $ClNO_2$ focused on investigating its influencing factors, such as the $N_2O_5$ uptake coefficient and the production yield of $ClNO_2$ (Thornton et al., 2003; Tham et al., 2018). The field and laboratory studies have indicated that $ClNO_2$ production was mainly affected by ambient temperature (T), relative humidity (RH), and particle components (e g., chloride ($Cl^-$), $NO_3^-$, and liquid



water content) (Bertram and Thornton, 2009; Wang et al., 2023; Wang et al., 2020). In addition to
influencing factors, the photochemical effects of $ClNO_2$ photolysis have been extensively evaluated (Xue et
al., 2015; Xia et al., 2021; Tham et al., 2016). Cl radical released by $ClNO_2$ photolysis will oxidize VOCs to
promote the formation $RO_2$ radical and $O_3$, greatly compensating for the underestimation of $RO_2$ radical and
$O_3$ generation in model simulations (Peng et al., 2021; Ma et al., 2023). The field measurements of $ClNO_2$
have been conducted in different atmospheric environments, while the key drivers of $ClNO_2$ chemistry were
still not well recognized. Moreover, it is pertinent to explore whether there are additional and unrecognized
sources of $ClNO_2$ beyond its heterogeneous generation from $N_2O_5$.
In this study, the comprehensive measurements of $ClNO_2$ and related parameters were conducted in a
coastal city of Southeast China during the autumn of 2022. Our research integrated field observations with
machine learning to reveal the key driving factors of $ClNO_2$ formation, particularly, investigating the
potential generation mechanisms of daytime $ClNO_2$. Additionally, we also assessed the photochemical
impacts of $ClNO_2$ based on a box model. Overall, this study underscored the important role of $NO_3^-$ in the
$ClNO_2$ chemistry.

**2 Materials and methods**
**2.1 Field Measurements**
The intensive field measurements of $ClNO_2$, related precursors, and meteorological parameters from
October 9th to December 5th, 2022 were performed at an urban site (Institute of Urban Environment,
Chinese Academy of Sciences) in a coastal city (Xiamen) of Southeast China (Fig. S1). Here, $ClNO_2$, $N_2O_5$,
gaseous pollutants (volatile organic compounds (VOCs), $NO_x$, $SO_2$, CO, and $O_3$), aerosol mass
concentrations, ionic components, size distribution, and meteorological factors were simultaneously detected.
Meanwhile, an iodide-adduct time-of-flight chemical ionization mass spectrometer ($I^-$-ToF-CIMS) was used
to measure $ClNO_2$ and $N_2O_5$. The principles and settings of $I^-$-ToF-CIMS were similar with previous studies
(Ma et al., 2023; Yan et al., 2023). Detailed descriptions of this observation site and instruments have been
provided in previous work (Chen et al., 2024; Hu et al., 2022), Text S1, and Table S1. For the calibrations of
$ClNO_2$ and $N_2O_5$, $ClNO_2$ was produced by passing $Cl_2$ (6 ppm in $N_2$) through a moist mixture of sodium
nitrite ($NaNO_2$) and sodium chloride (NaCl) (Thaler et al., 2011; Wang et al., 2022), and $N_2O_5$ was
synthesized by the reactions of $O_3$ and excessive $NO_2$ (Tham et al., 2016; Wang et al., 2016). The
dependences of $ClNO_2$ and $N_2O_5$ sensitivities on relative humidity are presented in Fig. S2. The details of
$ClNO_2$ and $N_2O_5$ calibrations and uncertainty analysis are displayed in Text S2.





**2.2 Machine Learning model**

Here, the extreme gradient boosting (XGBoost) model coupling with the Shapely additive explanations (SHAP) model (the XGBoost-SHAP model) was used to identify the key influencing factors of $ClNO_2$ formation. Meanwhile, the XGBoost model was applied to establish the predictive model of $ClNO_2$ based on the observed data of gaseous precursors and meteorological factors; the SHAP model was employed to evaluate the importance of each feature affecting the simulated concentrations of $ClNO_2$. Besides, the partial dependence plot (PDP) analysis offers a visual representation of the marginal effect that the factors have on the model's predicted outcome. It is based on the principle of stabilizing the values of non-target features, and systematically altered the target feature's values according to the model's algorithmic framework to derive the predicted values.

$ClNO_2$ concentrations served as the dependent variable, with trace gases ($SO_2$, $CO$, $NO_2$, $NO$, $O_3$, and $N_2O_5$), $PM_{2.5}$ and its inorganic compositions ($NO_3^-$, $SO_4^{2-}$, $NH_4^+$, and $Cl^-$), and meteorological parameters (T, RH, UV, WS, WD, and BLH) acting as independent variables. The simulated $ClNO_2$ concentrations by the XGBoost model were highly similar with the observed values ($R^2$=0.91), indicating the good performance of the XGBoost model (Fig. S3). Detailed introductions and settings of the XGBoost-SHAP model are provided in Text S3.

**2.3 The box model**

The observation-based model (OBM) was utilized to assess the impacts of $ClNO_2$ on photochemically atmospheric oxidation. As delineated in earlier studies (Xue et al., 2015; Tham et al., 2016; Xia et al., 2021; Peng et al., 2021; Peng et al., 2022), the Master Chemical Mechanism (MCM, version 3.3.1) was adopted, and established chlorine chemistry mechanisms have been integrated. The Tropospheric Ultraviolet and Visible Radiation (TUV) model was employed to determine the $ClNO_2$ photolysis rates ($JClNO_2$) under clear sky scenarios, subsequently calibrating $JClNO_2$ by measured $JNO_2$ values. A thorough exposition of the box model configuration can be found in our previous publications (Liu et al., 2022b; Liu et al., 2022a) and Text S4. Observation data, including $ClNO_2$, VOCs, HCHO, HONO, CO, $O_3$, NO, $NO_2$, $SO_2$, along with meteorological factors as constraint were input into the box model at an hourly resolution (Table S2). Two scenarios were examined: one representing observation-average conditions from October 9th to December 5th, the other reflecting a high $ClNO_2$ case observed on November 28th.

This study focused on elucidating the influence of $ClNO_2$ on the formation of $RO_x$ radical and $O_3$. The $O_3$ production rate minus the $O_3$ loss rate was used to calculate the net $O_3$ production rate (Eq. S1-3). The AOC is calculated by the sum of the rates of $CH_4$, CO, and VOCs oxidized by atmospheric oxidants ($O_3$, OH,





Cl, and $NO_3$ radical) (Eq. S4) (Xue et al., 2015; Yi et al., 2023). Both scenarios were evaluated with and
without including $ClNO_2$ inputs to assess its impacts on these processes.

**3 Results and discussion**
**3.1. Overview of observations**
Fig. 1 displays the time series of $ClNO_2$, $N_2O_5$, and related parameters including $O_3$, $NO_x$, $PM_{2.5}$, $Cl^-$,
$NO_3^-$, and meteorological parameters during the autumn observation period. Our observation shows a
decline in T and UV values from October to November, with average RH values increasing from ~ 60% in
October to ~ 70% in November (excluding rainy days). During the entire measurement period, $ClNO_2$
concentrations exhibited significant variability, with elevated levels (> 500 ppt) frequently observed in late
autumn, particularly after November 10th. The elevation of $ClNO_2$ concentrations coincided with increased
levels of $N_2O_5$ and $NO_3^-$ during late autumn. The concentrations of $ClNO_2$ at our study site reached several
ppb, compared with previous field measurements conducted at urban, suburban, rural, background, and
mountain sites (Table S3), indicating its widespread presence in diverse atmospheric environments. The
highest concentrations of $ClNO_2$ were detected at midnight of November 27th, with maximum hourly
average concentrations of 3.4 ppb. Simultaneously, peak concentrations of $N_2O_5$ and $NO_3^-$ were also
observed (Fig. 1). On the evening of November 27th, $N_2O_5$ concentrations rapidly decreased after 7 p.m.,
while $ClNO_2$ and $NO_3^-$ concentrations significantly increased, reflecting fast $N_2O_5$ heterogeneous hydrolysis
and effective formation of $ClNO_2$. Notably, on the following day (November 28th) (Fig. 2a), $ClNO_2$
concentrations sustained above 100 ppt around noon, partially related with weaken UV values (~ 14 $W \cdot m^{-2}$)
under heavy fog and cloud cover, with the RH values of ~ 70% at that time. Similar research in California
has shown $ClNO_2$ concentrations exceeding 100 ppt after sunrise 4 hours due to reduced photolysis (Mielke
et al., 2013).
The average diurnal changes of $ClNO_2$ and related parameters during the entire measurement campaign
were depicted in Fig. 2b. As expected, $ClNO_2$ exhibited a distinct diurnal variation, peaking and
accumulating after sunset and decreasing in the early morning. However, $ClNO_2$ concentrations remained ~
40 ppt around noon, different with some studies that $ClNO_2$ concentrations decreased to near the detection
limit around midday (Wang et al., 2022; Niu et al., 2022). $N_2O_5$ concentrations only presented a small peak
after sunset, and declined to near the detection limit in the daytime, suggesting minimal contribution from
daytime $ClNO_2$ formation via $N_2O_5$ heterogeneous uptake. Similar observation in North China declared
$ClNO_2$ concentrations above 60 ppt in the afternoon (Liu et al., 2017). Previous studies have indicated that





abundant $ClNO_2$ may be transported from upper atmosphere or air mass, contributing to the elevated $ClNO_2$
concentrations in the early morning (Tham et al., 2016; Xia et al., 2021; Jeong et al., 2019). However, the
explanations for the concentrations of $ClNO_2$ around noon remained elusive, implying additional sources
driving daytime $ClNO_2$ generation beyond $N_2O_5$ uptake.
**3.2. Key drivers of $ClNO_2$ formation**
The XGBoost-SHAP model was employed to investigate the major drivers of $ClNO_2$ production during
the whole observation period. The average absolute SHAP value of each feature was ranked to determine the
key drivers of $ClNO_2$ formation, with larger SHAP values suggesting greater contributions (Fig. 3a).
Additionally, features with positive SHAP values (depicted as red points) indicate that higher values of those
features positively affect $ClNO_2$ concentrations, and vice versa (Fig. 3b). Overall, $N_2O_5$, $NO_3^-$, T, RH, and
UV were the most important features affecting $ClNO_2$ concentrations. Notably, these factors exhibited varied
behaviors between daytime and nighttime periods (Fig. 5).
In our study, $N_2O_5$ was identified as the most important influencing factor, consistent with its role in
$ClNO_2$ formation through heterogeneous uptake processes (Thornton et al., 2010; Finlayson-Pitts et al.,
1989). After sunset, $ClNO_2$ concentrations markedly increased due to active nighttime $N_2O_5$ chemistry,
while this heterogeneous uptake process was hindered after sunrise as $N_2O_5$ concentrations decreased
significantly (Fig. 1) (Niu et al., 2022; Wang et al., 2020; Tan et al., 2022). Indeed, the concentrations of
$ClNO_2$ were evidently increased when $N_2O_5$ concentrations exceeded ~13 ppt, predominantly during the
nighttime (Fig. 4a). Conversely, in Northern Europe, the $ClNO_2$ concentrations were mainly controlled by
$O_3$ and $NO_2$, rather than by the heterogeneous uptake of $N_2O_5$ (Sommariva et al., 2018). In Heshan of South
China, chloride and $PM_{2.5}$ were the major factors affecting $ClNO_2$ formation (Wang et al., 2022). Differently,
$NO_3^-$ could play a vital role in affecting the concentrations of $ClNO_2$ alongside $N_2O_5$ in this study.
According to Fig. 4b, the high $NO_3^-$ concentrations (> 3.7 $\mu g \cdot m^{-3}$) corresponded to the elevation of $ClNO_2$,
especially its concentrations exceeding 6.2 $\mu g \cdot m^{-3}$. It is well recognized that $NO_3^-$ and $ClNO_2$ were co-
products from the processes of $N_2O_5$ heterogeneous uptake (Wang et al., 2017; Yun et al., 2018b). Hence, the
relative importance of $NO_3^-$ derived from the XGBoost-SHAP result indicated that the process of $NO_3^-$
formation is accompanied by the generation of $ClNO_2$ at night during our observation period. As mentioned
before, it is evidently observed that elevated concentrations of nighttime $ClNO_2$ were coincided with
increased $NO_3^-$ concentrations in late autumn. Considering the limited contribution of $N_2O_5$ hydrolysis to
daytime $NO_3^-$ levels (Yan et al., 2023; Zang et al., 2022; Chen et al., 2020), the impact of high $NO_3^-$
concentrations on daytime $ClNO_2$ concentrations warrants further analysis.





The simulated concentrations of $ClNO_2$, based on the XGBoost-SHAP model, were significantly
elevated when $NO_3^-$ concentrations were higher than 3.7 $\mu g \cdot m^{-3}$ (Fig. 4b). Consequently, the averagely daily
concentrations of $NO_3^-$ were classified as high (> 3.7 $\mu g \cdot m^{-3}$) and low (< 3.7 $\mu g \cdot m^{-3}$) to further elucidate the
impacts of $NO_3^-$ on the formation of $ClNO_2$. Fig. 5 presents the diurnal variations in the relative importance
of factors based on the SHAP values under high and low $NO_3^-$ concentrations. Unexpectedly, daytime $NO_3^-$
was the dominant influencing factors for daytime $ClNO_2$ (Fig. 5a). High concentrations of daytime $NO_3^-$
positively affected the daytime concentrations of $ClNO_2$, independent of $N_2O_5$ uptake processes. As depicted
in Fig. 5a, daytime $N_2O_5$ did not promoted the elevation of daytime $ClNO_2$. Therefore, it is very likely that
high concentrations of daytime $NO_3^-$ participated in daytime $ClNO_2$ production. A recent study declared that
nitrate photolysis produced $ClNO_2$ in addition to $Cl_2$ (Dalton et al., 2023), while it has been not verified by
field observations. Fig. 6 shows that daytime $ClNO_2$ concentrations corrected well (R=0.62) with the
product of a proxy of $NO_3^-$ photolysis ($NO_3^- \times J NO_2 \times S_a$) on aerosol surfaces ($S_a$), implying that the
photolysis of $NO_3^-$ contributed to the daytime concentrations of $ClNO_2$ at our study site. Furthermore, high
concentrations of $NO_3^-$ and $Cl^-$, along with large values of $S_a$ (Fig. 6a, b, c) in the daytime accelerated $NO_3^-$
photolysis, promoting the formation of $ClNO_2$. Overall, $N_2O_5$ uptake processes were the major pathways
dominating nighttime $ClNO_2$ formation, while $NO_3^-$ photolysis contributed to daytime $ClNO_2$ production
during our observation period.
In term of meteorological factors, UV, T, and RH were the major influencing factors. The photolysis
was the most important sink of $ClNO_2$ in the daytime, leading to a rapid reduction in $ClNO_2$ concentrations,
particularly in the early morning (Fig. 4e and Fig. 5). The weakened UV from October to November
decreased the photolysis rate of $ClNO_2$ (Fig. 1a), while $NO_3^-$ photolysis contributed partially to daytime
$ClNO_2$ concentrations (Fig. 6d), indicating the dual role of photolysis (or UV). The impact of T on $ClNO_2$
was probably reflected in its thermal equilibrium with $N_2O_5$. Elevated daytime T inhibited the formation of
$N_2O_5$ (Fig. 4c and Fig. 5). During the whole observation period from October to November, the drop in T
facilitated $ClNO_2$ production by decreasing the thermal decomposition process (Fig. 5). Increased RH values
provided favorable conditions for the nighttime $N_2O_5$ hydrolysis reactions, thereby affecting $ClNO_2$
production (Fig. 4d and Fig. 5), while high RH (> 80%) also weakened the generation of $ClNO_2$. Notably,
$Cl^-$ was not the most important factors of $ClNO_2$ formation at our study site (Fig. 3), likely attributed to the
abundant chlorine source in coastal regions (Peng et al., 2022).
**3.3. Photochemical effects of $ClNO_2$**
The photochemical effects of $ClNO_2$ were evaluated under the observation-average condition and the



high ClNO$_2$ case based on the box model. The largest Cl production rates (P(Cl)) contributed from ClNO$_2$
photolysis were 0.05 ppb·h$^{-1}$ for the observation-average condition, which was lower than 0.19 ppb·h$^{-1}$ for
the high ClNO$_2$ case. The difference led to variable levels of atmospheric oxidation capacity induced by Cl
radical. Cl radical released via the photolysis of ClNO$_2$ initiated the oxidation of VOCs. Among VOC
groups (including alkanes, alkenes, alkynes, aromatics and OVOCs), Cl radical primarily oxidized alkanes
(~ 65.0%), followed by OVOCs (~ 12.7%) for both the observation-average condition and the high ClNO$_2$
case (Fig. 7a, b). The contributions of Cl radical and other atmospheric oxidants (including OH radical and
O$_3$) to daytime VOC oxidation were also compared (Fig. 7c, d and Table 1). In our study, the oxidation of
alkanes by Cl radical for the observation-average condition were about 11.7%, which increased by 44.8%
for the high ClNO$_2$ case, were higher than those in London (Bannan et al., 2015), Weybourne (Bannan et al.,
2017), Boston (Rutherford et al., 1995), and LA (Fraser et al., 1997), lower than that in Hong Kong (Xue et
al., 2015). It should be noticed that the rates of Cl radical reacting with alkanes even exceeded those of OH
radical in the early morning for the high ClNO$_2$ case. The largest rates of alkanes oxidized by Cl radical were
approximately twice as high as those of OH radical at 10 a.m. (Fig. 7e, f), highlighting that the
photochemical effects of Cl radical released via ClNO$_2$ photolysis were particularly important for VOC
oxidation during the morning hours at our study site.
The oxidation of VOCs by Cl radical further affects the generation of RO$_x$ (OH + HO$_2$ + RO$_2$) radicals.
The RO$_x$ radical production rates for the high ClNO$_2$ case were evidently lower than that under the
observation-average condition, primarily due to reduced photolysis rates on that day. However, the total RO$_x$
radical production rates averagely increased by 23.8% with ClNO$_2$ photolysis for the high ClNO$_2$ case,
higher than a 4.9% increase for the observation-average condition (Fig. S4). For the observation-average
condition, O$_3$ (32.7%), HONO (31.7%), and OVOCs (21.6%) photolysis were the most significant
contributors to RO$_x$ radical production in the early morning (7-10 a.m.), with VOC oxidation by Cl radical
contributing only 3.7% (Fig. 8a). However, for the high ClNO$_2$ case, VOC oxidation induced by Cl radical
in the early morning accounted for 19.1% of RO$_x$ radical production, which was higher than O$_3$ (7.4%) and
HCHO (4.1%) photolysis, close to OVOCs (19.0%) photolysis (Fig. 8b). The contributions of ClNO$_2$
photolysis to the RO$_x$ radical production rates in our study were larger than previous results observed in
autumn of Heshan (Wang et al., 2022) and North China (Xia et al., 2021), similar with that in summer of
Wangdu (Tham et al., 2016). Thus, the concentrations of OH, HO$_2$, and RO$_2$ radicals in the box model with
ClNO$_2$ inputs averagely increased by 17.9%, 34.6%, and 54.3% for the high ClNO$_2$ case, higher than the
increases of 3.7%, 7.1%, and 10.3% contributed from the observation-average conditions, respectively (Fig.



S5). The uplift in the concentrations of $RO_x$ radicals also accelerated the generation of $O_3$. The increase in
the net $O_3$ production rates ($P(O_3)$) for the observation-average condition averagely reached 0.13 ppb·h$^{-1}$
(15.8 %) in the daytime (Fig. 9a), while larger elevations in the net $P(O_3)$ were observed for the high $ClNO_2$
case (Fig. 9b), with a maximum of 0.64 ppb·h$^{-1}$ (120 %) at 10 a.m. As a result, increased $RO_x$ radical and $O_3$
greatly enhanced the atmospheric oxidation capacity (Fig. 9c, d), especially for the high $ClNO_2$ case (up to

281    65%).

282        Table 2 summarizes the impacts of $ClNO_2$ photolysis on $RO_x$ radical and $O_3$ production in our study

and previous observations around the world (Xia et al., 2021; Wang et al., 2022; Tham et al., 2016; Wang et
al., 2016; Xue et al., 2015; Bannan et al., 2017; Jeong et al., 2019), indicating that the photochemical
impacts of $ClNO_2$ were variable in different atmospheric environments. At our study site, the effects of
$ClNO_2$ photolysis on $RO_x$ radical production were important, especially in the early morning. The enhanced
$RO_x$ radical production induced by $ClNO_2$ photolysis accelerated the chemical generation of $O_3$. Primary
$RO_x$ radical production rates (including $O_3$, HONO, HCHO, OVOCs, and $ClNO_2$) were considered as one of
the most important parameters to $O_3$ formation (Lu et al., 2023). Therefore, the considerable contribution of
$ClNO_2$ photolysis to primary $RO_x$ radical production in the early morning may bring new challenges for $O_3$
alleviation.

**Conclusions**

294        In conclusion, we present two months of field measurements in the coastal area of Southern China

during the autumn, coupled with machine learning and model simulations, providing new insights into
$ClNO_2$ chemistry. Our observation shows the increase in the concentrations of $ClNO_2$ were accompanied by
elevated concentrations of $N_2O_5$ and $NO_3^-$, low values of T and UV, and high values of RH. The nighttime
heterogeneous uptake of $N_2O_5$ was identified as the major source of $ClNO_2$, while $NO_3^-$ photolysis
promoted the elevation of daytime $ClNO_2$ concentrations. Cl radical released by $ClNO_2$ photolysis after
sunrise had important photochemical effects in the early morning. The photolysis of high $ClNO_2$
concentrations resulted in net $O_3$ production rates and atmospheric oxidation capacity levels increasing by
120% and 65%, respectively. Our results enhanced the understanding of $ClNO_2$ chemistry in coastal regions,
calling for more observations and laboratory research to fully reveal its exact role in different atmospheric
environments.

**Data availability.** Data are available upon request to Jinsheng Chen (jschen@iue.ac.cn).




**Author contributions.** JC provided funding support for field measurements, designed this study, and revised this manuscript. GC designed this study, analyzed the data, and wrote this manuscript. HW helped perform the calibrations and revised this manuscript. XF revised this manuscript. XF, HW, YT, ZL, XJ, LX, BH contributed to discussions of this manuscript.

**Competing interests.** The authors declare that they have no conflict of interest.

**Acknowledgements.** The authors acknowledge the National Natural Science Foundation of China, the Science and Technology Department of Fujian Province, Center for Excellence in Regional Atmospheric Environment Project, Xiamen Atmospheric Environment Observation and Research Station of Fujian Province, and Fujian Key Laboratory of Atmospheric Ozone Pollution Prevention (Institute of Urban Environment, Chinese Academy of Sciences).

**Financial support.** This work was funded by the National Natural Science Foundation of China (U22A20578, 42305102 & 42277091), the Science and Technology Department of Fujian Province (2022L3025), the National Key Research and Development Program (2022YFC3700304), STS Plan Supporting Project of the Chinese Academy of Sciences in Fujian Province (2023T3013), Fujian Provincial Environmental Protection Science & Technology Plan Projects (2023R004), and Xiamen Atmospheric Environment Observation and Research Station of Fujian Province. Y.J.T. acknowledges the funding support from the Guangdong Basic and Applied Basic Research Foundation (2022A1515010852) and the Fundamental Research Funds for the Central Universities, Sun Yat-sen University (23hytd002).

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

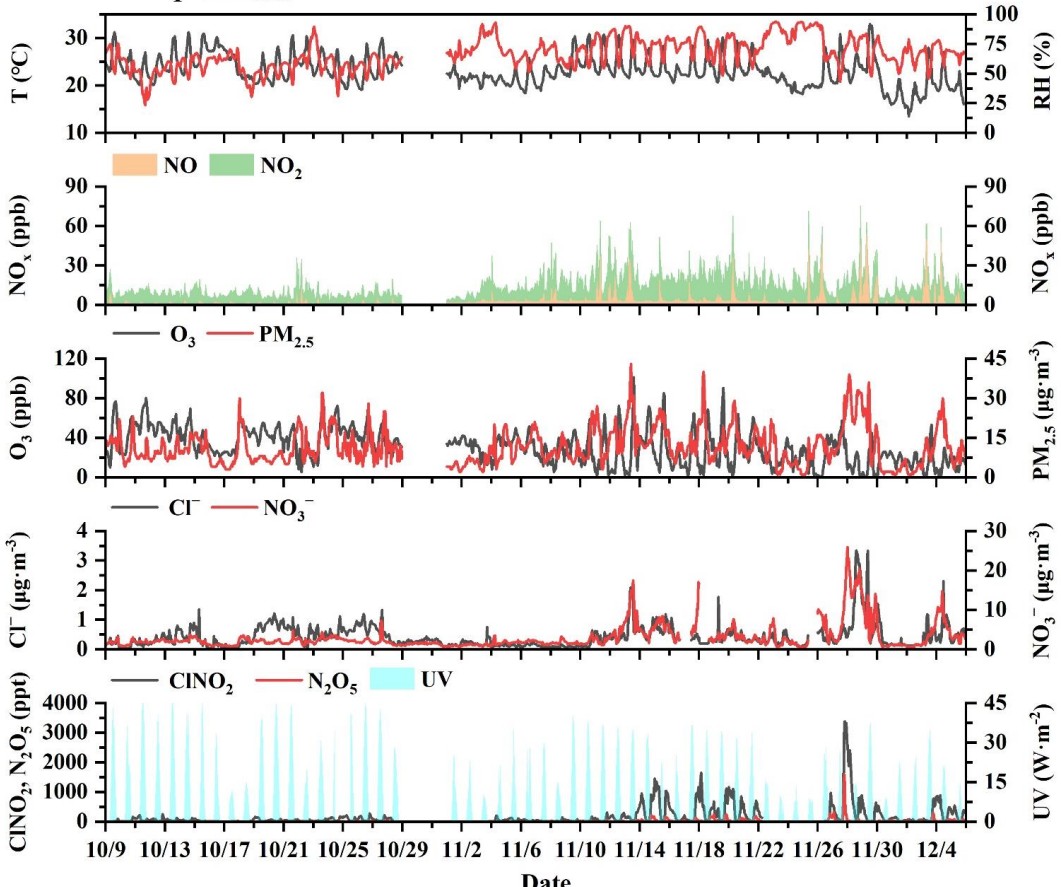


Figure 1. The time series of ClNO₂, related precursors, and meteorological parameters during the autumn
observation period.



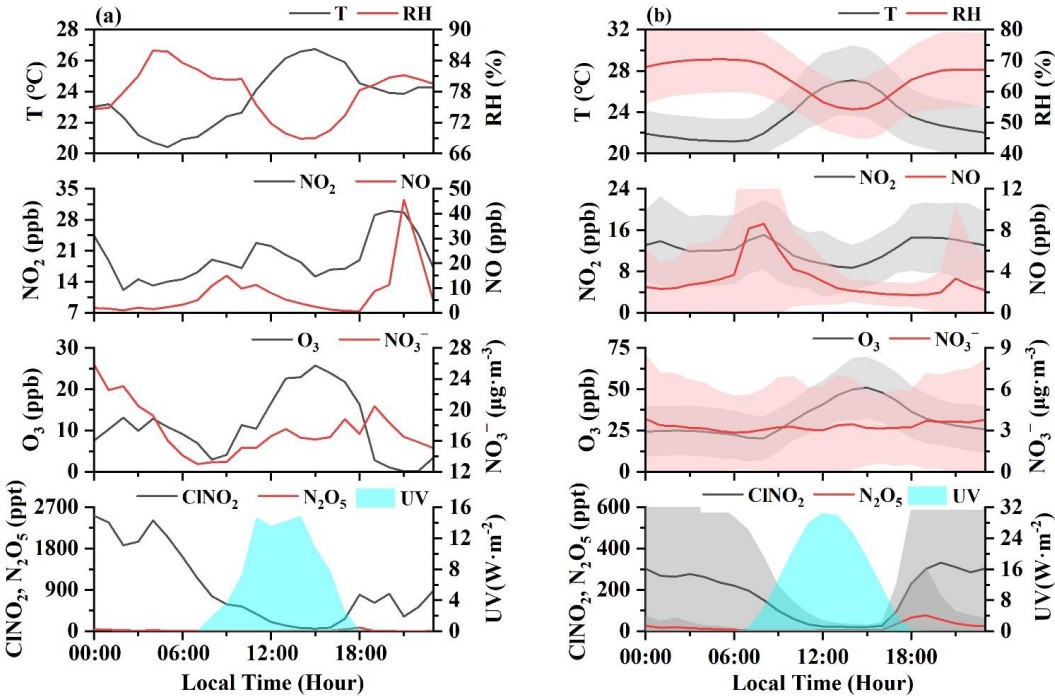

Figure 2. Diurnal variations of ClNO$_2$ and other related parameters for the highest concentrations of ClNO$_2$

(case) on November 28th (a) and the observation-average condition (from 9 October to 5 December) (b).



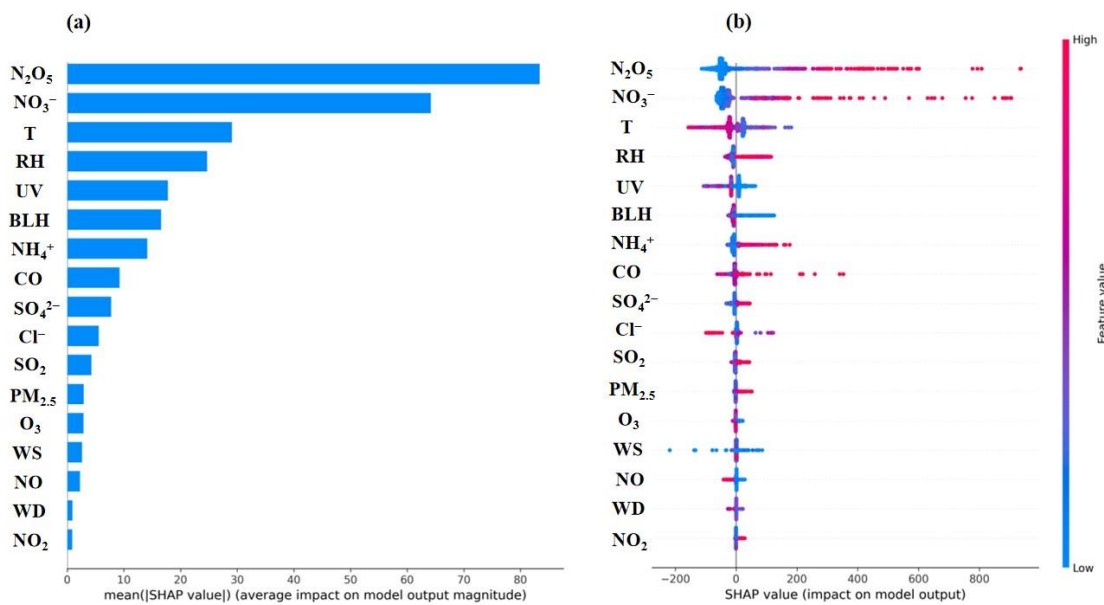

Figure 3. Relative importance of each feature to ClNO$_2$ using XGBoost-SHAP during the autumn observation period. The mean absolute SHAP value (a), summary plot of SHAP values of each feature (b).



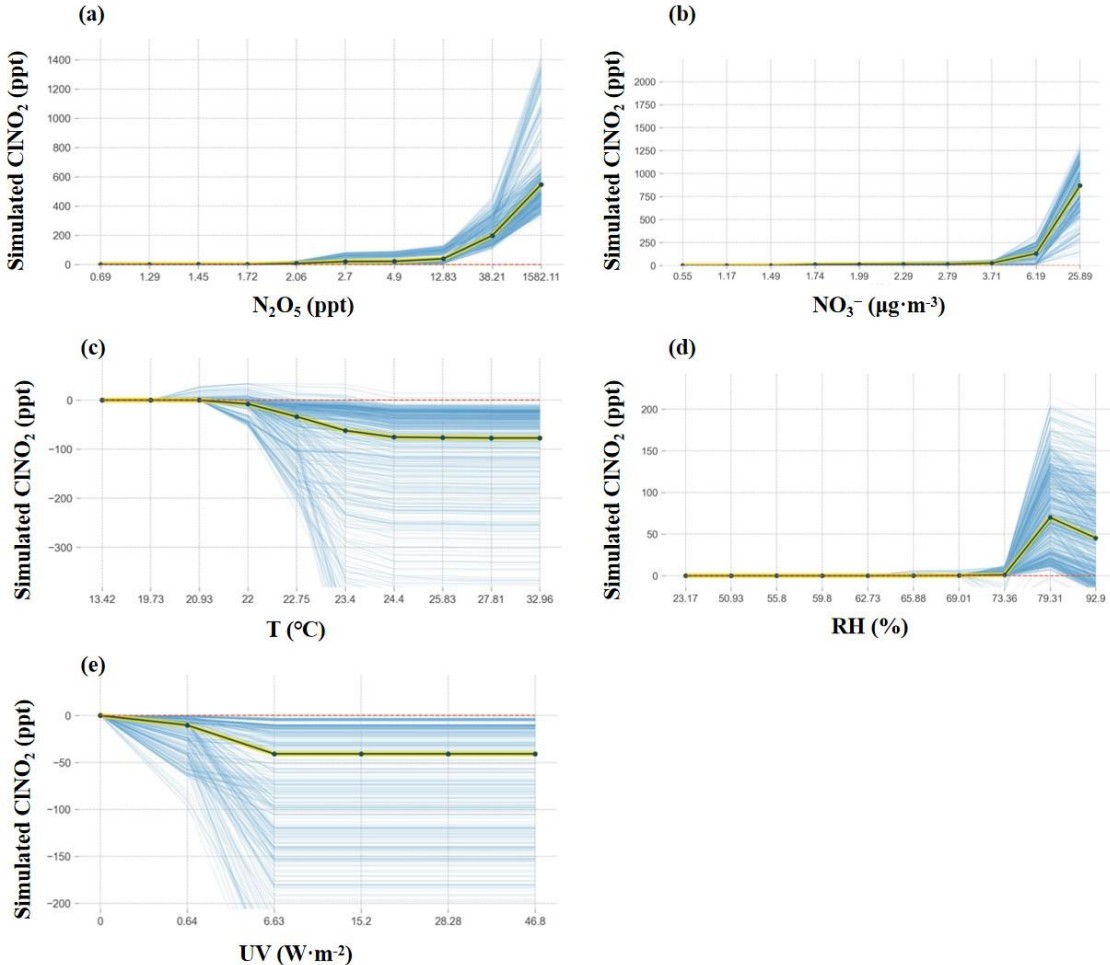

Figure 4. Isolation plots of PDP for $N_2O_5$ (a), $NO_3^-$ (b), T (c), RH (d), and UV (e). The average variations of simulated $ClNO_2$ with factors' changes spline are indicated by the yellow and black curve, and blue curves presents all situations during the whole observation period.






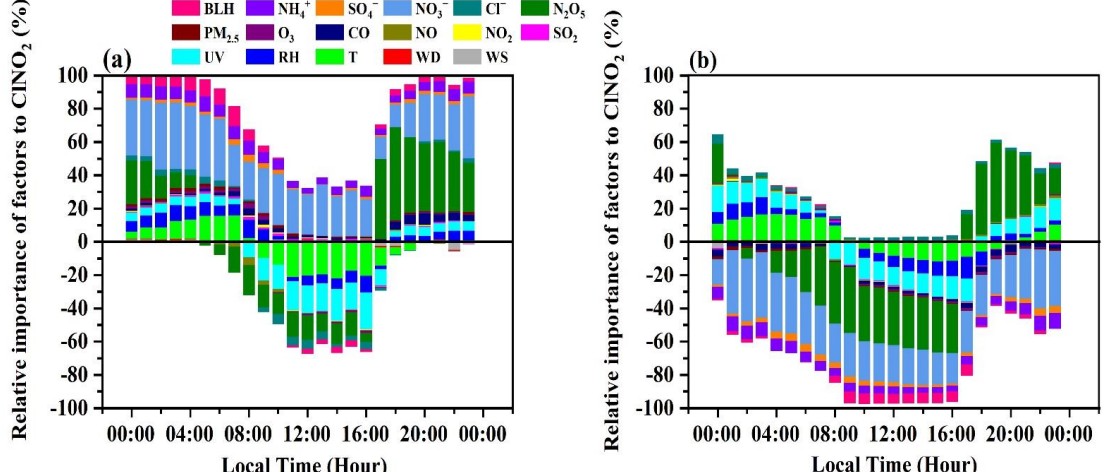


Figure 5. The diurnal variations of the relative importance of factors to ClNO$_2$ based on the SHAP values
under the high (> 3.7 μg·m$^{-3}$) (a) and low (< 3.7 μg·m$^{-3}$) (b) ClNO$_2$ concentrations.







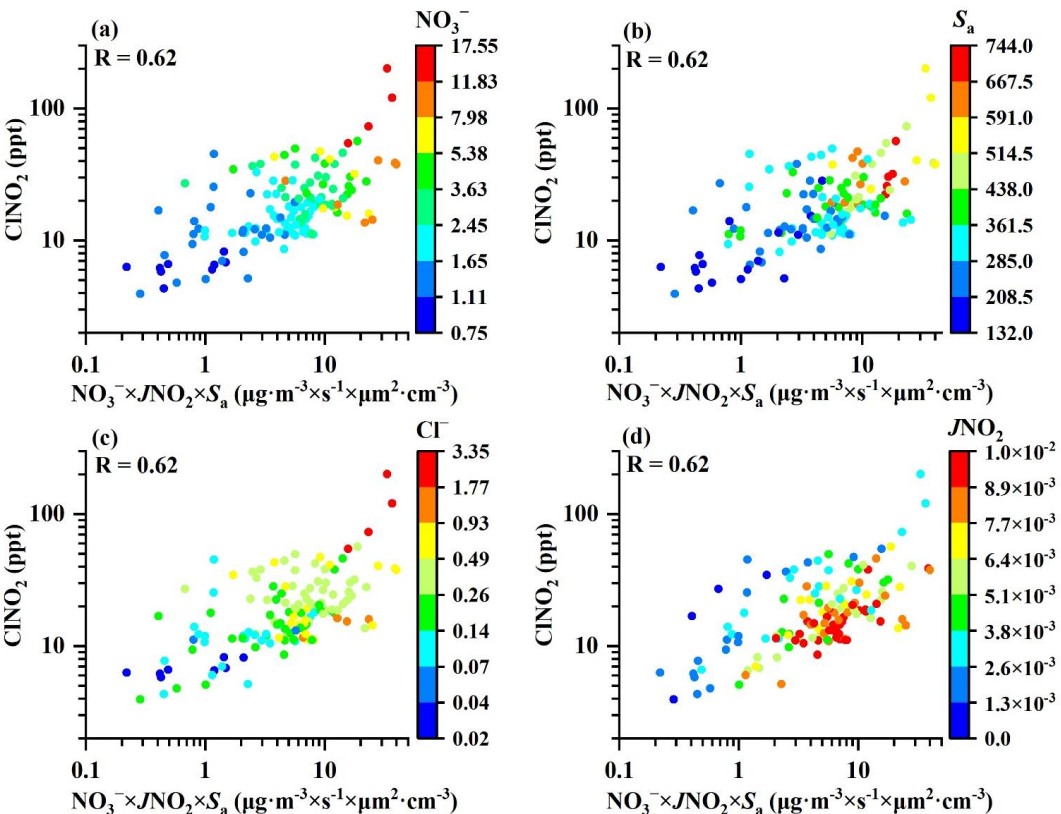

Figure 6. The relationship of daytime $ClNO_2$ concentrations (12:00-15:00 Local Time) and a proxy of nitrate ($NO_3^-$) photolysis ($NO_3^- \times JNO_2 \times S_a$). The color of the dots respects the $NO_3^-$ (a), $S_a$ (b), $Cl^-$ (c), $JNO_2$ (d), respectively.





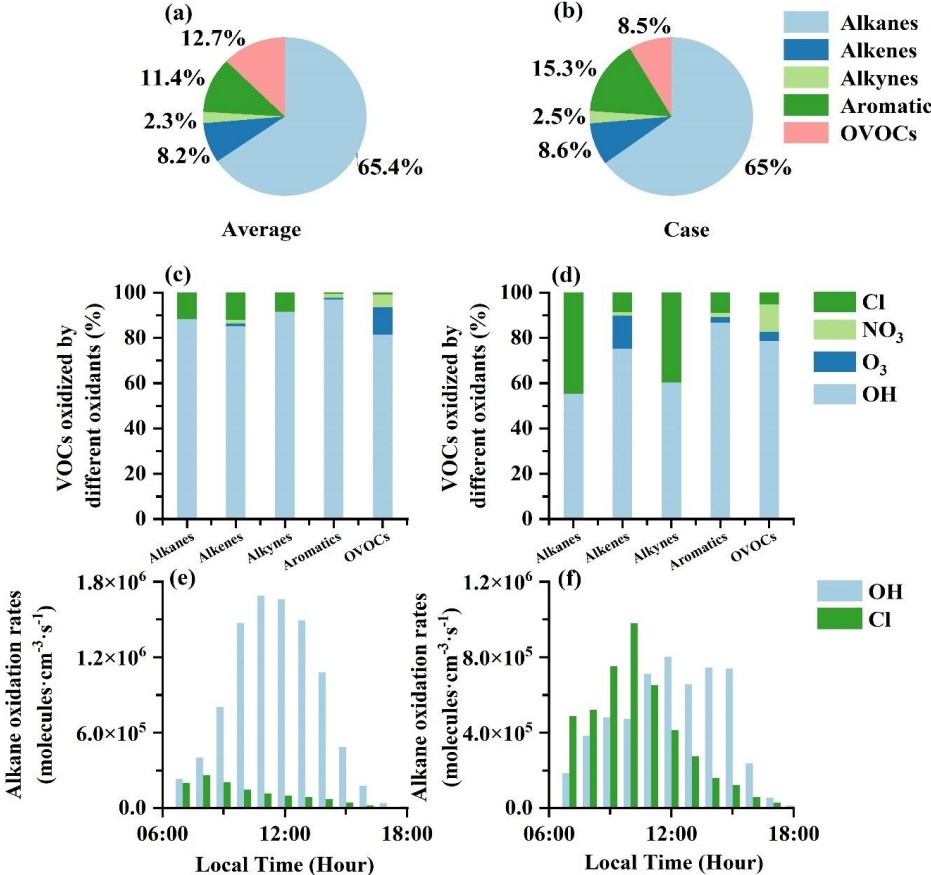

Figure 7. The impacts of Cl radials released by ClNO$_2$ photolysis and other atmospheric oxidants (including OH, NO$_3$, and O$_3$) on VOC oxidation under the observation-average condition and high ClNO$_2$ case, respectively. The contributions of different VOC groups oxidized by Cl radical during the observation-average (a). The contributions of different VOC groups oxidized by Cl radical during the case (b). The contributions of different atmospheric oxidants (including OH, Cl, NO$_3$, and O$_3$) to VOC groups during the observation-average (c). The contributions of different atmospheric oxidants (including OH, Cl, NO$_3$, and O$_3$) to VOC groups during the case (d). Comparisons of alkane oxidation rates (molecules·cm$^{-3}$·s$^{-1}$) by OH and Cl radical during the observation-average (e). Comparisons of alkane oxidation rates by OH and Cl radical (molecules·cm$^{-3}$·s$^{-1}$) during the case (f).





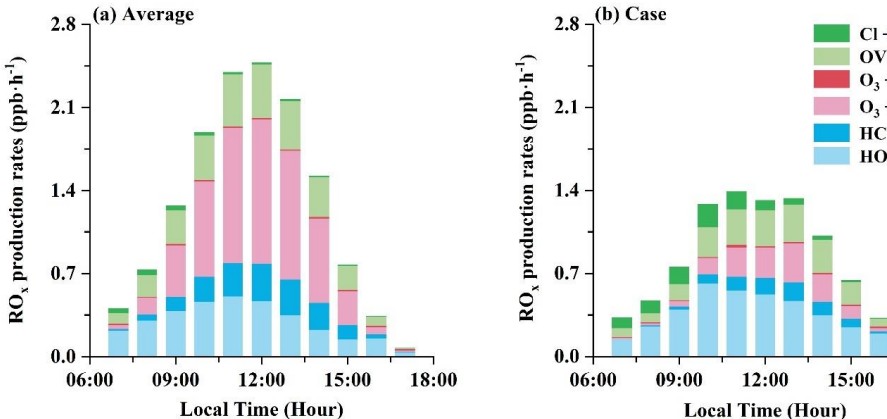

Figure 8. The contributions of different production pathways to $RO_x$ production rates under the observation-average condition (a) and high $ClNO_2$ case (b), respectively.





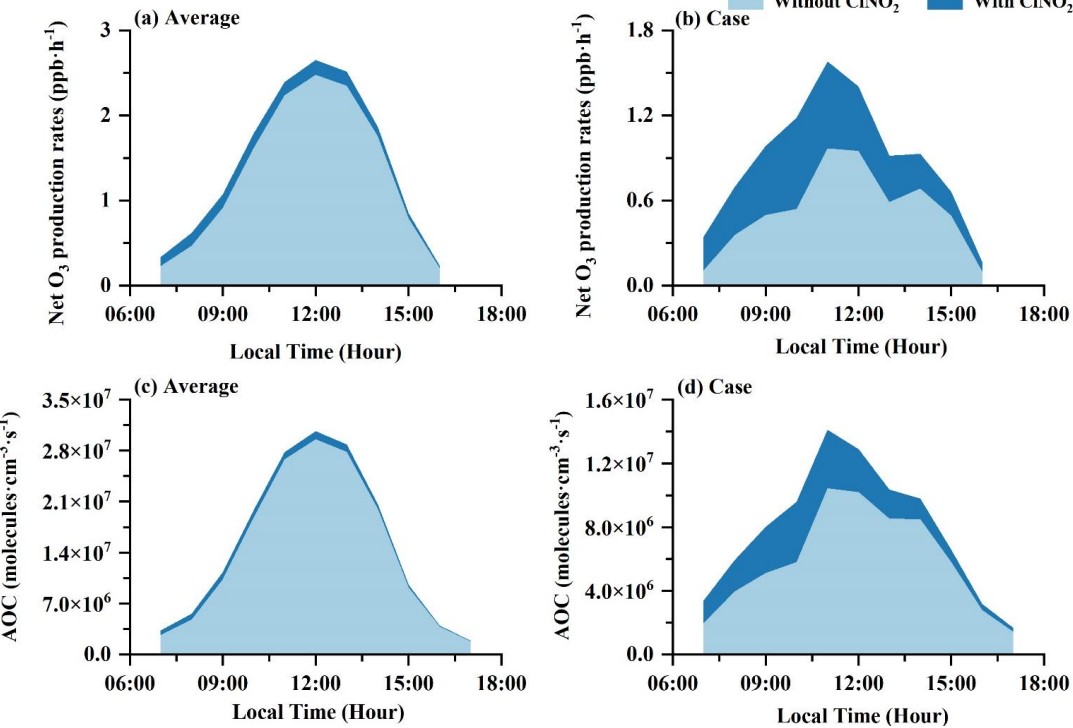


Figure 9. The impacts of Cl radials released by ClNO$_2$ photolysis on net O$_3$ production rates and the AOC
levels under the observation-average condition (a, c) and high ClNO$_2$ case (b, d), respectively.

















Table 1. Relative importance of Cl, OH, and $O_3$ to the daytime oxidation of VOC groups (including alkanes,
alkenes, alkynes, aromatics, and OVOCs) around the world (Xue et al., 2015; Bannan et al., 2015; Bannan et
al., 2017; Rutherford et al., 1995; Fraser et al., 1997).

| | Xiamen (average) | Xiamen (case) | Hong Kong (max) | London (average) | Weybourne (average) | Boston | LA |
|---|---|---|---|---|---|---|---|
| Alkane Cl% | 11.7 | 44.8 | 53.0 | 3.5 | 1.0 | 8.5 | 9.9 |
| Alkane OH% | 88.3 | 55.2 | 47.0 | 96.5 | 99.0 | 91.5 | 90.1 |
| Alkane $O_3$% | - | - | - | - | - | - | - |
| Alkene Cl% | 12.2 | 8.7 | 14.0 | 0.6 | 0.4 | 0.3 | 0.3 |
| Alkene OH% | 85.0 | 75.2 | 81.0 | 77.9 | 78.3 | 33 | 31.3 |
| Alkene $O_3$% | 1.2 | 14.7 | 5.0 | 21.5 | 21.4 | 66.7 | 68.4 |
| Alkyne Cl% | 8.5 | 40.0 | - | 7.0 | 2.6 | 8.7 | 8.7 |
| Alkyne OH% | 91.5 | 60.0 | - | 91.8 | 96.7 | 89.7 | 89.7 |
| Alkyne $O_3$% | - | - | - | 1.2 | 0.7 | 1.6 | 1.6 |
| Aromatics Cl% | 0.7 | 9.1 | 11.0 | - | - | - | - |
| Aromatics OH% | 97.0 | 86.6 | 89.0 | - | - | - | - |
| Aromatics $O_3$% | 0.7 | 2.6 | - | - | - | - | - |
| OVOCs Cl% | 0.9 | 5.2 | 6.0 | - | - | - | - |
| OVOCs OH% | 81.4 | 78.7 | 85.0 | - | - | - | - |
| OVOCs $O_3$% | 12.0 | 3.9 | - | - | - | - | - |


















Table 2. The impacts of $ClNO_2$ photolysis on $RO_x$ (OH, $HO_2$, and $RO_2$) levels, $P(RO_x)$, and $P(O_3)$ around the
world (Xia et al., 2021; Wang et al., 2022; Tham et al., 2016; Wang et al., 2016; Xue et al., 2015; Bannan et
al., 2017; Jeong et al., 2019).

| Study Area | Season | OH | HO₂ | RO₂ | P(RO_x) | P(O₃) |
|---|---|---|---|---|---|---|
| Xiamen (average) | Autumn | 3.7% | 7.1% | 10.3% | 4.9% | 6.7% |
| Xiamen (case) | Autumn | 17.9% | 34.6% | 54.3% | 23.8% | 41.7% |
| Wangdu/Beijing/Mt. Tai | Winter | 15.0%–22.0% | 24.0%–31.0% | 36.0%–52.0% | 1.3%–3.8% | 1.3%–6.2% |
| Heshan | Autumn | 1.5%–2.6% | 1.9%–4.6% | 3.0%–6.8% | < 2.2% | 1.0%–4.9% |
| Wangdu | Summer | - | - | - | 10%–30% | 3.0%–13.0% |
| Mt. Tai Mo Shan, Hong Kong | Winter | 40.0%–77.0% | 53.0%–106.0% | - | - | 11.0%–41.0% |
| Hok Tsui, Hong Kong | Summer | 6.6% | 12.2% | 45.1% | - | 10.3% |
| Weybourne | Spring | 5.0% | 7.0% | 9.0% | - | - |
| Seoul | Spring | - | - | - | - | 1.0%–2.0% |
