# Peer review of "Formation drivers and photochemical effects of ClNO2 in a coastal city of"

_EGUsphere, 2024_

## Author Comment (AC1)

**Responses to Editors and Reviewers**

We appreciate the reviewers for the constructive and helpful comments, the incorporation of which has led to an improved manuscript. We have revised the manuscript appropriately and addressed the reviewer's comments in our point-by-point responses. As detailed below, the reviewer's comments are shown in black, our responses to the comments are in blue fonts, and the added/rewritten parts are presented in red fonts.

**RC1**: **'Comment on egusphere-2024-1638'**, **Anonymous Referee #1, 31 Jul 2024**

**General Comments**

The work by Gaojie Chen et al. is a well written study presenting two months of ambient observations in Southeast China and has two main components. First, the work introduces interesting evidence for the formation of $ClNO_2$ during the daytime by a recently suggested particulate nitrate mechanism. Second, the work discusses the implications for Cl radical production from $ClNO_2$ photolysis.

The first component has significant implications for the understanding of $ClNO_2$ formation globally. However, a discussion of the traditional metrics of $ClNO_2$ formation, the $N_2O_5$ uptake rate and $ClNO_2$ yield, are completely absent from the paper. Without a discussion on this topic, the authors' conclusion that "$NO_3^-$ photolysis contributed to daytime generation" is severely weakened. In fact, it is based only a machine learning output which gauges the "importance" of $NO_3^-$ influence on $ClNO_2$ as well as a linear regression of $ClNO_2$ with $NO_3^-$ $\times jNO_2 \times$ aerosol $S_a$. In this joint correlation, insufficient evidence is provided to suggest that the photolysis component improves the correlation. As such, I request major revisions in which the authors justify their conclusion by demonstrating that the daytime observations of $ClNO_2$ cannot be explained by traditional $N_2O_5$ and $ClNO_2$ chemistry.

The second component is based on box modeling from the master chemical mechanism. Aside from a lack of detail on the parametrization used for $N_2O_5$ uptake and

ClNO$_2$ yield, the results presented are generally sound and informative. I request that the authors include their choice of parametrization in the main text.

**Response:** Thanks for your valuable comments. Your review comments and suggestions are benefit to improve the quality and readability of this manuscript. We have revised the manuscript appropriately and addressed the reviewer's comments point-by-point for consideration as below.

The first component: We have added the discussions on the N$_2$O$_5$ uptake coefficient ($\gamma$(N$_2$O$_5$)) and ClNO$_2$ yield ($\phi$(ClNO$_2$)). Furthermore, we also provided the evidence showing that the daytime observations of ClNO$_2$ cannot be explained by traditional N$_2$O$_5$ and ClNO$_2$ chemistry. Please refer to our response to Specific Comment 4 for more details.

The second component: In this study, the box model is employed to evaluate the photochemical effects of ClNO$_2$. The levels of ClNO$_2$ in the box model were constrained by the observed levels of ClNO$_2$ from our field measurements. This approach eliminates the necessity for parameterization of N$_2$O$_5$ uptake and ClNO$_2$ yield to determine ClNO$_2$ levels. Therefore, the parametrization for N$_2$O$_5$ uptake and ClNO$_2$ yield was not utilized in the box model.

**Specific Comments**

1. Section 2: A description on the handling of N$_2$O$_5$ uptake and ClNO$_2$ yield is absent from the methods. A list of previous papers is provided but it is not clear how these two parameters are handled. Both N$_2$O$_5$ uptake and ClNO$_2$ yield will vary with the parameters investigated here (T, RH, etc.). See McDuffie et al.

McDuffie, E. E., Fibiger, D. L., Dubé, W. P., Lopez Hilfiker, F., Lee, B. H., Jaeglé, L., et al. (2018a). ClNO$_2$ yields from aircraft measurements during the 2015 WINTER campaign and critical evaluation of the current parameterization. Journal of Geophysical Research: Atmospheres, 123(22), 12994–13015. https://doi.org/10.1029/2018JD029358

McDuffie, E. E., Fibiger, D. L., Dubé, W. P., Lopez-Hilfiker, F., Lee, B. H., Thornton, J. A., et al. (2018b). Heterogeneous N$_2$O$_5$ uptake during winter: Aircraft measurements

during the 2015 WINTER campaign and critical evaluation of current parameterizations. Journal of Geophysical Research: Atmospheres, 123(8), 4345–4372. https://doi.org/10.1002/2018JD028336

**Response:** Thanks for your comment. In this study, the box model is employed to evaluate the photochemical effects of $ClNO_2$. The levels of $ClNO_2$ in the box model were constrained by observed levels of $ClNO_2$ from our field measurements. This approach negates the need for parameterization of $N_2O_5$ uptake and $ClNO_2$ yield to determine $ClNO_2$ levels. Therefore, the parametrization for $N_2O_5$ uptake and $ClNO_2$ yield was not utilized in the box model.

**Added/rewritten:** "Due to the levels of $ClNO_2$ in the box model determined by observed levels of $ClNO_2$, the parametrization for $N_2O_5$ uptake and $ClNO_2$ yield was not utilized in the box model."

2. Section 3.1: There is no uncertainty presented with the observations in the main text. Please include the uncertainties as the uncertainties in the SI are non-negligible (~20 %).

**Response:** Thanks for your comment. We have included the uncertainties in the main text.

**Added/rewritten:** "The uncertainties of the $ClNO_2$ and $N_2O_5$ measurements were estimated to be ~20 %."

3. Figure 5: What is the interpretation of negative "importance factors"? During the daytime, $N_2O_5$ is a negative importance factor. Please discuss this in the main text.

**Response:** Thanks for your comment. In the XGBoost-SHAP model, SHAP values are used to quantify the contribution of each feature to the prediction values, with a negative SHAP value indicating a negative contribution. Generally, negative "importance factors" suggest that the presence of these factors contributes minimally or decreases the predicted values of the dependent variable. Therefore, in our study, negative SHAP values for $N_2O_5$ during the daytime indicate that the contribution of $N_2O_5$ chemistry to daytime $ClNO_2$ levels was limited. We have added these discussions.

**Added/rewritten:** "Generally, negative "importance factors" suggest that the

presence of these factors contributes minimally or decreases the predicted values of the dependent variable. Therefore, in our study, negative SHAP values for $N_2O_5$ during the daytime indicate that the contribution of $N_2O_5$ chemistry to daytime $ClNO_2$ levels was limited."

4. Section 3.2: A discussion on the changes in aerosol content (particulate nitrate) and the effect on $N_2O_5$ uptake and $ClNO_2$ yield is absent. Such a discussion is critical here. Traditionally, one expects nitrate to reduce $N_2O_5$ uptake (the nitrate effect) which would limit the production of $ClNO_2$. Even so, $ClNO_2$ could be enhanced in a high nitrate case if the $N_2O_5$ uptake and $ClNO_2$ yield are substantially greater than low nitrate air masses. According to Figure 1, there are concurrent enhancements of pCl and $pNO_3$ during some time periods. As pCl increases the $ClNO_2$ yield will also increase which would then be (coincidentally?) concurrent with high $pNO_3$ Even more, these periods of concurrent pCl and $pNO_3$ appear to correlate with enhanced $PM_{2.5}$ and thus, I assume, aerosol surface area. Increases in surface area would then increase $N_2O_5$ uptake further promoting $ClNO_2$ and $pNO_3$ production. Lastly, Figure 6 suggests that the correlation between $ClNO_2$ mixing ratio and $pNO_3xjNO_2xSa$ is driven by $pNO_3xSa$ while $jNO_2$ has a limited or no correlation (panel d). In other words, photolysis appears to have a limited role in the production of $ClNO_2$.

While the above may be speculative, it is an example of why a lack of discussion on the $ClNO_2$ yield and $N_2O_5$ uptake significantly weakens the arguments made by the authors. As written, I believe there is insufficient evidence to conclude that "$NO_3^-$ photolysis contributed to daytime [$ClNO_2$] generation".

**Response:** Thanks for your valuable comments. The $N_2O_5$ uptake coefficient ($\gamma(N_2O_5)$) and $ClNO_2$ yield ($\phi(ClNO_2)$) were estimated using the observational data and parameterization. We derived the values of and $\phi(ClNO_2)$ based on increased rates of $ClNO_2$ and particle nitrate ($NO_3^-$) in the field observation (Phillips et al., 2016). Specially, $\gamma(N_2O_5)$ and $\phi(ClNO_2)$ were calculated by Eq. (1) and (2).

$$\gamma(N_2O_5) = \frac{2 \times \left( P(ClNO_2) + P(NO_3^-) \right)}{cN_2O_5 S_a [N_2O_5]} \qquad (1)$$

$$\phi(\text{ClNO}_2) \;=\; 2 \times \left(1 \;+\; \frac{P(NO_3^-)}{P(\text{ClNO}_2)}\right)^{-1} \tag{2}$$

Here, $P(\text{ClNO}_2)$ and $P(NO_3^-)$ represent the production rates of $\text{ClNO}_2$ and $NO_3^-$ induced by $N_2O_5$ uptake, respectively. $S_a$ denotes the aerosol surface area, and $c(N_2O_5)$ is the mean molecular speed of $N_2O_5$. This method assumes that air masses remain relatively stable, and $\text{ClNO}_2$ and $NO_3^-$ were produced through nighttime $N_2O_5$ heterogeneous uptake. More details on the method are provided elsewhere (Tham et al., 2018; Niu et al., 2022; Phillips et al., 2016). Using the method and selection criteria, we derived $\gamma(N_2O_5)$ and $\phi(\text{ClNO}_2)$ during the whole measurement period.

[Figure]

Figure 1. The relationship between field-derived $\gamma(N_2O_5)$ (a), $\phi(\text{ClNO}_2)$ (b) and $NO_3^-$ concentrations.

The relative importance of $NO_3^-$ derived from the XGBoost-SHAP result indicated that $NO_3^-$ could play a vital role in affecting the concentrations of $\text{ClNO}_2$ besides $N_2O_5$. The high $NO_3^-$ concentrations ($> 3.7$ $\mu g \cdot m^{-3}$) are accompanied by the elevation of $\text{ClNO}_2$, especially its concentrations reaching 6.2 $\mu g \cdot m^{-3}$. Previous studies declared that the increased concentrations of $NO_3^-$ decreased $\gamma(N_2O_5)$, which would limit the production of $\text{ClNO}_2$ (Wahner et al., 1998; Mentel et al., 1999; Bertram and Thornton, 2009). As depicted in Figure 1, the dependence of $\gamma(N_2O_5)$ on $NO_3^-$ concentrations follows the nitrate suppression effect, which subsequently hindered further $\text{ClNO}_2$ formation. Therefore, the importance of nighttime $NO_3^-$ for $\text{ClNO}_2$ levels is that they are co-products from the processes of $N_2O_5$ heterogeneous uptake. During our filed

observation, compared to low $NO_3^-$ conditions, $ClNO_2$ production was enhanced in high $NO_3^-$ conditions. Especially in late autumn, the increased aerosol surface area and $N_2O_5$ levels enhanced $N_2O_5$ uptake, which further promoted both $ClNO_2$ and $NO_3^-$ production.

To evaluate the contribution of the heterogeneous $N_2O_5$ uptake to daytime $ClNO_2$ levels, we calculated $ClNO_2$ production using Eq. (3), considering the loss of $ClNO_2$ through photolysis. This method has been employed in a previous study (Tham et al., 2016).

$$\frac{d[ClNO_2]}{dt} = k(N_2O_5)[N_2O_5]\phi(ClNO_2) - JClNO_2[ClNO_2] \qquad (3)$$

$$k(N_2O_5) = \frac{1}{4}cN_2O_5S_a\gamma(N_2O_5) \qquad (4)$$

We used a $\gamma(N_2O_5)$ value of 0.06 and a $\phi(ClNO_2)$ value of 1.0 in our calculations, which represented upper-end estimates based on previous field studies (Mcduffie et al., 2018a; Mcduffie et al., 2018b; Tham et al., 2016). However, as shown in Figure 2, a $\phi(ClNO_2)$ of 1.0 with a $\gamma(N_2O_5)$ of 0.06 ($\phi\gamma = 0.06$) fails to reproduce the observed levels of daytime $ClNO_2$. A larger $\gamma(N_2O_5)$ of 0.11 would be necessary, but such high uptake coefficients and yields are not supported by the current literature. Therefore, we believe that the observed daytime $ClNO_2$ levels, particularly around noon, cannot be adequately explained by heterogeneous $N_2O_5$ uptake alone, suggesting the presence of additional sources contributing to the formation of daytime $ClNO_2$.

Notably, the laboratory research had confirmed that $NO_3^-$ photolysis can produce $ClNO_2$ (Dalton et al., 2023). In our study, machine learning analysis, which gauges the "importance" of $NO_3^-$ in affecting daytime $ClNO_2$, as well as a linear regression of $ClNO_2$ against $NO_3^- \times JNO_2 \times S_a$, implied that $NO_3^-$ photolysis contributed to daytime $ClNO_2$ concentrations at our study site. Although $NO_3^-$ photolysis can produce $ClNO_2$, this does not necessarily mean that higher photolysis intensity will result in higher $ClNO_2$ concentrations. It is crucial to understand the dual role of photolysis intensity in determining daytime $ClNO_2$ levels. Photolysis can contribute to the generation of $ClNO_2$ by promoting $NO_3^-$ photolysis, while also causing the rapid decomposition of $ClNO_2$. As reported in California (Mielke et al., 2013), reduced photolysis rates even

increased daytime ClNO$_2$ levels by decreasing ClNO$_2$ loss through photolysis. Additionally, in real atmospheric conditions, several factors beyond photolysis influence NO$_3^-$ photolysis, including NO$_3^-$ concentrations and particulate chloride levels.

[Figure]

Figure 2. Comparisons of daytime ClNO$_2$ levels between observation, and calculation using Eq. (4) with a $\phi$(ClNO$_2$) of 1.0 and a $\gamma$(N$_2$O$_5$) of 0.06 ($\phi\gamma$ = 0.06), or a $\phi$(ClNO$_2$) of 1.0 and a $\gamma$(N$_2$O$_5$) of 0.11 ($\phi\gamma$ = 0.11).

**Added/rewritten:** "N$_2$O$_5$ concentrations only presented a small peak after sunset, and declined to near the detection limit in the daytime. Previous studies indicated that abundant ClNO$_2$ could be transport from upper atmosphere or air mass, which contributed to ClNO$_2$ concentrations in the early morning (Tham et al., 2016; Xia et al., 2021; Jeong et al., 2019). However, the explanations for the concentrations of ClNO$_2$ around noon remained elusive. To evaluate the contribution of the heterogeneous N$_2$O$_5$ uptake to daytime ClNO$_2$ levels, we calculated ClNO$_2$ production using Eq. (3), considering the loss of ClNO$_2$ through photolysis. This method has been employed in a previous study (Tham et al., 2016).

We used a $\gamma(N_2O_5)$ value of 0.06 and a $\phi(ClNO_2)$ value of 1.0 in our calculations, which represent upper-end estimates based on previous field studies (Mcduffie et al., 2018a; Mcduffie et al., 2018b; Tham et al., 2016). However, as shown in Figure. 2R, a $\phi(ClNO_2)$ of 1.0 with a $\gamma(N_2O_5)$ of 0.06 ($\phi\gamma = 0.06$) fails to reproduce the observed levels of daytime $ClNO_2$. A larger $\gamma(N_2O_5)$ of 0.11 would be necessary, but such high uptake coefficients and yields are not supported by the current literature. Therefore, we believe that the observed daytime $ClNO_2$ levels, particularly around noon, cannot be adequately explained by heterogeneous $N_2O_5$ uptake alone, suggesting the presence of additional sources contributing to the formation of daytime $ClNO_2$."

"The relative importance of $NO_3^-$ derived from the XGBoost-SHAP result indicated that $NO_3^-$ could play a vital role in affecting the concentrations of $ClNO_2$ besides $N_2O_5$. Moreover, according to Figure. 4b, the high $NO_3^-$ concentrations ($> 3.7$ $\mu g \cdot m^{-3}$) are accompanied by the elevation of $ClNO_2$, especially its concentrations reaching 6.2 $\mu g \cdot m^{-3}$. Previous studies declared that increased concentrations of $NO_3^-$ decreased $\gamma(N_2O_5)$, which would limit the production of $ClNO_2$ (Wahner et al., 1998; Mentel et al., 1999; Bertram and Thornton, 2009). As depicted in Figure 1, the dependence of $\gamma(N_2O_5)$ on $NO_3^-$ concentrations follows the nitrate suppression effect, which subsequently hindered further $ClNO_2$ formation. Therefore, the importance of nighttime $NO_3^-$ for $ClNO_2$ levels is that they are co-products from the processes of $N_2O_5$ heterogeneous uptake. During our field observation, compared to low $NO_3^-$ conditions, $ClNO_2$ production was enhanced in high $NO_3^-$ conditions. Especially in late autumn, the increased aerosol surface area and $N_2O_5$ levels enhanced $N_2O_5$ uptake, which further promoted both $ClNO_2$ and $NO_3^-$ production."

"It is crucial to understand the dual role of photolysis intensity in determining daytime $ClNO_2$ levels. Photolysis can contribute to the generation of $ClNO_2$ by promoting $NO_3^-$ photolysis, while also causing the rapid decomposition of $ClNO_2$. As reported in California (Mielke et al., 2013), reduced photolysis rates even increased daytime $ClNO_2$ levels by decreasing $ClNO_2$ loss through photolysis."

**Technical Comments**

Line 76: tenths: tens

**Response:** Thanks for your comment. We have revised it.

**Added/rewritten:** "Since Osthoff et al. (2008) firstly detected over 1 ppb of ClNO$_2$ in the urban outflows of America, significant production of ClNO$_2$ was widely observed in the polluted coastal and inland areas with abundant anthropogenic emissions and chloride sources, and its concentrations were ranged from tens of ppt to several ppb."

Figure 3, 5 and 6: Please change the color scale to a colorblind friendly version.

**Response:** Thanks for your comment. We have changed the color scale in Figure 3, 5 and 6 to a colorblind friendly version. Additionally, due to N$_2$O$_5$, NO$_3^-$, T, RH, and UV being the most important features of affecting ClNO$_2$ concentrations, we only compared their relative importance. Therefore, Figure 5 only presents the relative importance of N$_2$O$_5$, NO$_3^-$, T, RH, and UV.

**Added/rewritten:**

[Figure]

Figure 3. Relative importance of each feature to ClNO$_2$ using XGBoost-SHAP during the autumn observation period. The mean absolute SHAP value (a), summary plot of SHAP values of each feature (b).

[Figure]

Figure 5. The diurnal variations of the relative importance of the major five factors (including $N_2O_5$, $NO_3^-$, T, RH, and UV) to $ClNO_2$ based on the SHAP values under the high (> 3.7 μg·m$^{-3}$) (a) and low (< 3.7 μg·m$^{-3}$) (b) $ClNO_2$ concentrations.

[revised manuscript text omitted]

---

## Author Comment (AC2)

**Responses to Editors and Reviewers**

We sincerely appreciate the reviewers for their constructive and insightful comments, which are of great benefit to improve the quality of the manuscript. In response, we have carefully revised the manuscript and addressed each comment in a point-by-point manner. For clarity, the reviewers' comments are presented in black, our responses in blue, and the added or revised sections of the manuscript are highlighted in red.

**RC2**: 'Comment on egusphere-2024-1638', Anonymous Referee #3, 03 Jan 2025

In this manuscript, the authors present a study that investigates key factors driving the production of $ClNO_2$ based on field observations and XGBoost-SHAP model. Furthermore, the authors evaluated the potential impact of $ClNO_2$ photolysis on the formation of $RO_2$ and hence, the atmospheric oxidative capacity.

Overall, I found this manuscript interesting and well-constructed. Although the conclusion drawn for the nighttime $ClNO_2$ formation has been well recognized for two decades, the contribution of $NO_3^-$ photolysis to daytime $ClNO_2$ is confirmed by the authors, which brings sufficient novelty to this manuscript.

Despite this, I do have some comments, particularly on the interpretation of the machine learning results, which need to be fully addressed before this manuscript can be accepted for publication.

**Response:** Thank you for your valuable and thoughtful comments. Your comments and suggestions have greatly enhanced the overall quality and readability of the manuscript. We have made the necessary revisions and provided detailed responses to each point below for your consideration.

**General comments:**

1. Machine learning, especially SHAP value, starts to be widely used in atmospheric research very recently, but many readers may not be sufficiently familiar with it. To improve the readability, I believe the way of interpreting SHAP values must be fully informed in the manuscript. E.g., what do the negative and positive SHAP values

stand for? Should the contribution be evaluated by the true value or absolute value.

**Response:** Thank you for your comment. We have added a detailed introduction to SHAP values in the revised manuscript.

**Added/rewritten:** "The SHAP model is an interpretability tool designed to analyze the contributions of individual features to model predictions. It employs an additive explanatory framework that considers all features as contributors, drawing inspiration from cooperative game theory. For each predicted instance, SHAP assigns a Shapley value, representing the cumulative contribution of each feature. Positive SHAP values indicate that a feature increases the model's predicted outcome, signifying a positive contribution. Conversely, negative SHAP values suggest that the feature reduces the predicted value, reflecting a negative contribution. The absolute value of the SHAP score reflects the magnitude of the contribution, regardless of direction, offering insight into the overall importance of the feature. The true value, on the other hand, reveals the direction of the contribution (positive or negative), facilitating a clearer understanding of the relationship between the feature and the prediction."

2. I am not fully convinced by the way of performing SHAP model and its interpretation.

1) why does the aerosol surface, as a known important factor for $N_2O_5$ uptake, not used as an input of SHAP model?

**Response:** Thank you for your valuable comment. We agree that aerosol surface area is a crucial factor influencing the heterogeneous uptake of $N_2O_5$. Initially, we had included particle surface area concentrations ($S_a$) in the XGBoost-SHAP model to assess its significance in $ClNO_2$ formation. However, the results indicated that $S_a$ did not play a prominent role (Figure R1). Furthermore, it is found that $R^2$ values of the training and testing sets slightly improved from 0.963 and 0.861 to 0.965 and 0.891, respectively, when $S_a$ was not used as an input of a machine learning model. Given that $PM_{2.5}$ and its inorganic compositions serve as representative indicators of aerosol conditions to some extent, we chose not to include aerosol surface area as a dependent variable in the machine learning model

to avoid redundancy.

[Figure]

Figure R1. Relative importance of each feature to ClNO₂ using the XGBoost-SHAP model during the autumn observation period, with $S_a$ included as an additional variable in the model.

2) ClNO₂ has a rather long nighttime lifetime, which means ClNO₂ could be accumulated during airmass transport. Meanwhile, N₂O₅ could both form and loss through the transport, leading to varying patterns of its concentration. In fact, this can be testified by calculating the maximal ClNO₂ production through N₂O₅ uptake by, e.g., assuming gamma = 0.1 and ClNO₂ yield = 1. Given this assumption, I didn't see any model input that could represent the influence of airmass transport. I suggest to reconsider their model input and incorporate certain transport parameters.

**Response:** Thank you for your thoughtful comment. I fully agree with your opinion that ClNO₂ tends to accumulate at night. We had indeed considered the impact of air mass transport in our analysis. In this study, trace gases (SO₂, CO, NO₂, NO, O₃, and N₂O₅), PM₂.₅ and its inorganic compositions (NO₃⁻, SO₄²⁻, NH₄⁺, and Cl⁻), along with meteorological parameters (T, RH, UV, WS, WD, and BLH)

were selected as independent variables. Typically, WS and WD effectively reflect the influence of air masses and play a significant role in the transport, dispersion, and accumulation of atmospheric pollutants. However, results from the XGBoost-SHAP model indicate that WS and WD have a minimal impact on $ClNO_2$ concentrations (Figure R2). Notably, previous observations indicating that $ClNO_2$ is easily influenced by air mass transport were primarily conducted in clean rural areas or under background atmospheric conditions (Niu et al., 2022; Tan et al., 2022). Given that our study site located in a typical urban area surrounded by shopping malls, residential zones, and major traffic arteries, it is highly affected by fresh anthropogenic emissions. Therefore, these results suggest that $ClNO_2$ concentrations are primarily driven by local processes, rather than by air mass transport during our study period.

[Figure]

Figure R2. Relative importance of each feature to $ClNO_2$ using the XGBoost-SHAP model during the autumn observation period.

3) As this study suggested, daytime and nighttime $ClNO_2$ are driven by different processes, which however, were affected by similar parameters (in different ways). For instance, $NO_3^-$ is a co-product with $ClNO_2$ at nighttime, but a precursor of

ClNO$_2$ in the daytime. I suggest to consider conducting SHAP models daytime and nighttime data sets separately, so that the exact role of these parameters can be better revealed.

**Response:** Thanks for your constructive comment. We fully agree with your insightful perspective. Through our in-depth analysis, we found that ClNO$_2$ exhibits distinctly different influence pathways during the daytime and nighttime, with certain parameters potentially playing different roles in these two periods. To investigate this further, we integrated all daytime and nighttime data into a unified machine learning model, resulting in a high-performing model. Using SHAP analysis, we were able to effectively distinguish the roles of key influencing factors between daytime and nighttime.

While the primary formation mechanisms of ClNO$_2$ differ between daytime and nighttime, there is a clear interconnection between daytime and nighttime ClNO$_2$ concentrations. Especially, the elevated nighttime ClNO$_2$ concentrations can significantly affect its concentrations in the early morning. Machine learning models trained exclusively on daytime data show poor performance, with R$^2$ values for the testing sets dropping below 0.6, thereby constraining further analysis of factor importance. As a result, separating daytime and nighttime data for independent machine learning analyses may risk overlooking the intrinsic linkages between these periods.

We believe that a comprehensive analysis, incorporating both daytime and nighttime data, is crucial for a complete and accurate assessment of ClNO$_2$ production and loss processes. Although we did not segregate the data into daytime and nighttime subsets for machine learning, SHAP analysis enabled us to clearly identify the relative importance of various factors during the daytime and nighttime, providing deeper insights into their respective mechanisms across these two periods.

For example, we used SHAP analysis to evaluate the key influencing factors of daytime ClNO$_2$. The simulated concentrations of ClNO$_2$, based on the XGBoost-SHAP model, were significantly elevated when NO$_3^-$ concentrations were higher than 3.7 μg·m$^{-3}$. Consequently, the average daily concentrations of NO$_3^-$ were classified as high (> 3.7 μg·m$^{-3}$) and low (< 3.7 μg·m$^{-3}$) to further elucidate the impacts of NO$_3^-$ on the

formation of ClNO₂. Fig. R3 presents the diurnal variations in the relative importance of the most critical influencing factors based on the SHAP values under high and low $NO_3^-$ concentrations. Unexpectedly, daytime $NO_3^-$ was the dominant influencing factors for daytime ClNO₂ (Fig. R3a). High concentrations of daytime $NO_3^-$ positively affected the daytime concentrations of ClNO₂, independent of $N_2O_5$ uptake processes. As depicted in Fig. R3a, daytime $N_2O_5$ did not promote the elevation of daytime ClNO₂. Negative SHAP values for $N_2O_5$ during the daytime indicate that the contribution of $N_2O_5$ chemistry to daytime ClNO₂ levels was limited. Therefore, it is very likely that high concentrations of daytime $NO_3^-$ participated in daytime ClNO₂ production.

[Figure]

Figure R3. The diurnal variations of the relative importance of factors to ClNO₂ based on the SHAP values under the high (> 3.7 μg·m⁻³) (a) and low (< 3.7 μg·m⁻³) (b) ClNO₂ concentrations.

**Detailed comments:**

Line 64 "were" could be replaced by "are", as this is common case.

> **Response:** Thanks for your comment. We have revised it.

> **Added/rewritten:** "The reaction rates between Cl radical and some alkanes are several orders of magnitude faster than those involving OH radical."

Line 99-100 "our research integrated…." This sentence has grammatic error, please rephrase.

> **Response:** Thanks for your comment. This sentence has been rephrased.

> **Added/rewritten:** "Field observations, combined with a machine learning model, were used to reveal the key driving factors of $ClNO_2$ formation. Furthermore, we further investigated the potential mechanisms driving daytime $ClNO_2$ generation."

Line 141-143. The statement of $JClNO_2$ calculation is not clear, please consider to rephrase.

> **Response:** Thanks for your comment. The statement of $JClNO_2$ calculation has been rephrased.

> **Added/rewritten:** "The Tropospheric Ultraviolet and Visible Radiation (TUV) model was used to calculate $ClNO_2$ photolysis rates ($JClNO_2$) under clear-sky conditions. The simulated $JClNO_2$ values were then scaled based on field-measured $JNO_2$ values."

Line 167-168 "Simultaneously, …" I think the high correlation between $ClNO_2$ and $N_2O_5$ (and $NO_3^-$) does not mean simultaneous peaking. From Fig.1, I can clearly see that their concentrations do not reach the maxima at exactly the same time.

> **Response:** Thanks for your valuable comment. We agree with your opinion that the concentrations of $ClNO_2$, $N_2O_5$, and $NO_3^-$ did not reach their maxima simultaneously. We intended to convey that their peak concentrations were observed during the night of November 27th. The sentences have been revised accordingly.

**Added/rewritten:** "The highest concentrations of $ClNO_2$ were detected during the night of November 27th, with a maximum hourly average of 3.4 ppb. Peak concentrations of $N_2O_5$ and $NO_3^-$ were also observed on that night."

Line 203-204 the authors first indicate $NO_3^-$ could affect the formation of $ClNO_2$; but afterwards, the authors say that the high $NO_3^-$ and $ClNO_2$ together were caused by the simultaneous formation. Please improve the logic of this part.

**Response:** Thanks for your comment. We have improved the logic of this part.

**Added/rewritten:** "Differently, the relative importance of $NO_3^-$ derived from the XGBoost-SHAP result indicated that elevated $ClNO_2$ concentrations were associated with high concentrations of $NO_3^-$ besides $N_2O_5$. According to Fig. 5b, high $NO_3^-$ concentrations ($> 3.7$ $\mu g \cdot m^{-3}$) are accompanied by the elevation of $ClNO_2$, especially its concentrations reaching 6.2 $\mu g \cdot m^{-3}$. The importance of nighttime $NO_3^-$ for $ClNO_2$ levels is that they are co-products from the processes of $N_2O_5$ heterogeneous uptake. As shown in Fig. 1, compared to low $NO_3^-$ conditions, $ClNO_2$ production was enhanced in high $NO_3^-$ conditions."

Line 221 "did not promoted…" should be "did not promote".

**Response:** Thanks for your comment. We have revised it.

**Added/rewritten:** "As depicted in Fig. 5a, daytime $N_2O_5$ did not promote the elevation of daytime $ClNO_2$."

Line 222 "A recent study declared that…". Please use "suggested" or "argued" instead of "declared".

**Response:** Thanks for your comment. We have revised it.

**Added/rewritten:** "A recent study suggested that nitrate photolysis produced $ClNO_2$ in addition to $Cl_2$ (Dalton et al., 2023), while it has been not verified by field observations."

Line 236-237. I am not convinced by the discussion about the role of temperature. The

authors suggested that $N_2O_5$ is not important for $ClNO_2$ in the daytime. Then how can temperature affect $ClNO_2$ through the thermal equilibrium of $N_2O_5$? Also, $N_2O_5$ is a measured quantity. Such a temperature impact should be already reflected by the connection between daytime $N_2O_5$ and $ClNO_2$.

**Response:** Thank you for your comments. We believe that $N_2O_5$ plays a critical role in the formation of $ClNO_2$, as $ClNO_2$ is generated through the heterogeneous uptake of $N_2O_5$ on chloride-containing aerosols. In this study, we emphasized that limited contribution of heterogeneous $N_2O_5$ uptake to daytime $ClNO_2$ concentrations was primarily due to very low daytime $N_2O_5$ levels, which are largely associated with its thermal decomposition. In other words, the thermal decomposition process affects $ClNO_2$ generation by reducing the availability of $N_2O_5$ in the daytime. Specifically, the elevated ambient temperature from nighttime to daytime reduced $N_2O_5$ concentrations through enhanced thermal decomposition. During the entire observation period from October to November, the overall drop in ambient temperature facilitated $ClNO_2$ production by reducing the thermal decomposition of $N_2O_5$, thereby increasing its availability for heterogeneous uptake.

**Added/rewritten:** "The impact of ambient temperature on $ClNO_2$ was probably reflected in its thermal equilibrium with $N_2O_5$. Elevated daytime ambient temperature suppressed the formation of $N_2O_5$, resulting in low $N_2O_5$ concentrations, which further limited the contribution of heterogeneous $N_2O_5$ uptake to daytime $ClNO_2$ generation. During the whole observation period from October to November, the drop in ambient temperature facilitated $ClNO_2$ production by decreasing the thermal decomposition process."

Line 243 I suggest the subtitle of "Impact of $ClNO_2$ photolysis on $RO_x$ budget"

**Response:** Thanks for your suggestion. We have revised it.

**Added/rewritten:** "3.3 Impact of $ClNO_2$ photolysis on $RO_x$ budget."

Figure 2: the $N_2O_5$ in the lowest panel is barely seen. Please consider to show the pattern by perhaps $N_2O_5*5$.

**Response:** Thank you for your suggestion. We have revised Figure 2 to update the presentation of $N_2O_5$ accordingly.

**Added/rewritten:**

[Figure]

Figure 2. Diurnal variations of $ClNO_2$ and other related parameters for the highest concentrations of $ClNO_2$ (case) on November 28th (a) and the observation-average condition (from 9 October to 5 December) (b).

Figure 4. the division of x ticks looks strange. Please modify.

**Response:** Thanks for your comment. We have modified Figure 4.

**Added/rewritten:**

[Figure]

Figure 4. Isolation plots of PDP for $N_2O_5$ (a), $NO_3^-$ (b), T (c), RH (d), and UV (e). The average variations of simulated $ClNO_2$ with factors' changes spline are indicated by the yellow and black curve, and blue curves presents all situations during the whole observation period.

**References**

Dalton, E. Z., Hoffmann, E. H., Schaefer, T., Tilgner, A., Herrmann, H., and Raff, J. D.: Daytime Atmospheric Halogen Cycling through Aqueous-Phase Oxygen Atom Chemistry, J. Am. Chem. Soc., 145, 15652-15657, https://doi.org/10.1021/jacs.3c03112, 2023.

Niu, Y.-B., Zhu, B., He, L.-Y., Wang, Z., Lin, X.-Y., Tang, M.-X., and Huang, X.-F.: Fast Nocturnal Heterogeneous Chemistry in a Coastal Background Atmosphere and Its Implications for Daytime Photochemistry, J. Geophys. Res. Atmos., 127, e2022JD036716, https://doi.org/10.1029/2022JD036716, 2022.

Tan, Z., Fuchs, H., Hofzumahaus, A., Bloss, W. J., Bohn, B., Cho, C., Hohaus, T., Holland, F., Lakshmisha, C., Liu, L., Monks, P. S., Novelli, A., Niether, D., Rohrer, F.,

Tillmann, R., Valkenburg, T. S. E., Vardhan, V., Kiendler-Scharr, A., Wahner, A., and Sommariva, R.: Seasonal variation in nitryl chloride and its relation to gas-phase precursors during the JULIAC campaign in Germany, Atmos. Chem. Phys., 22, 13137-13152, https://doi.org/10.5194/acp-22-13137-2022, 2022.

---

## Referee Report (RR1)

I appreciate the response from Gaojie Chen et al. to the reviewer responses. However, the main concerns indicated by reviewer 1 are still apparent.

**Comment 1:**

Referring to the original reviewer 1 comment 4: the authors have responded by providing a traditional analysis of ClNO2 production. This analysis is very helpful in gauging the potential for formation of ClNO2 from other mechanisms.

While the analysis seems appropriate the author's conclusion in the reply is not justified :"we believe that the observed daytime ClNO2 levels, particularly around noon, cannot be adequately explained by heterogeneous N2O5 uptake alone, suggesting the presence of additional sources contributing to the formation of daytime ClNO2."

The authors provided Figure R2 and argue that Figure R2 shows that observations of ClNO2 cannot be explained by an upper-limit calculation of N2O5 uptake ($\gamma$) and ClNO2 yield ($\phi$). In other words, the observed ClNO2 is greater than what could be produced by traditional chemistry. **Yet, the observation uncertainty is absent from the figure.**

**The authors have also mis-quoted their measurement uncertainty.** Reviewer 1 requested the authors state their uncertainty in the main text (comment 2) but the reviewer said it is "non-negligible (~20%)". The authors copied this number to the main text. However, this is incorrect. The SI includes the sensitivities: "The sensitivities of ClNO2 and N2O5 were 0.055 ± 0.018 and 0.11 ± 0.063 ncps·ppb−1" (lines 119-120). This translates to an uncertainty of 33% for ClNO2 and 57% for N2O5.

As such, the author's calculation of ClNO2 in their response and Figure R2 would have an observed uncertainty which combines both uncertainties of ClNO2 and N2O5 since the calculation of ClNO2 involves both compounds (equation 3 in author's response). These uncertainties should be added in quadrature as a result.

$$uncertainty = \sqrt{33\%^2 + 57\%^2} = 83\%$$

Remaking Figure R2 (presented below as Figure R2a) shows that the two upper-limit scenarios provided by the authors do indeed fall within the measurement uncertainty.

[Figure]

Figure R2a: reproduced from author's response. The data for plotting was acquired by the "Igor Thief" tool in Igor Pro by Wavemetrics. The only addition is the black error bars showing an 83% uncertainty on the observations.

It is understood that the cases presented by the authors represent upper-limit estimations. However, Figure R2a above shows that the $\phi\gamma$ = 0.06 case could produce results which match the observations within the uncertainty. I agree that the $\phi\gamma$ = 0.06 case is closer to an upper-limit estimation but I argue it is still within the realm of possibility (see references already provided by reviewer 1). Further, additional cases with lower $\phi\gamma$ values will very likely fall within the observational uncertainty.

My interpretation of the author's results is that their observations could be explained by traditional N2O5 chemistry. While there is precedent to consider the particulate nitrate mechanism, there is not clear evidence here to say "NO3− photolysis promoted the elevation of daytime ClNO2 concentrations" (line 327). This conclusion and others similar to it should be dampened to say that nitrate photolysis may have contributed but cannot be confirmed. The author's Figure R2 should also be included in the SI with included observational errors.

**Comment 2:**

The authors did not address concerns regarding the correlation of NO3-×jNO2×Sa. I understand that reviewer 1 did not explicitly request a change here. However, the point that panel 7d appears to show little correlation with jNO2 still remains. This point was never commented on by the authors. Without showing separated correlations between all factors it is impossible to know the drivers of this apparent correlation. As such the author's conclusion ("the photolysis of NO3− contributed to the daytime concentrations of ClNO2 at our study site") cannot be made.